# Long-Term Impacts of Selective Logging on Amazon Forest Dynamics from Multi-Temporal Airborne LiDAR

**Ekena Rangel Pinagé [1,2,](#)** ![ID], **Michael Keller [3,4,5](#)** ![ID], **Paul Duffy [6](#), Marcos Longo [4,5](#),**
**Maiza Nara dos-Santos [5](#) and Douglas C. Morton [7](#)**

1   School of Life Sciences, University of Technology Sydney, Sydney, NSW 2007, Australia
2   College of Forestry, Oregon State University, Corvallis, OR 97331, USA
3   USDA Forest Service, International Institute of Tropical Forestry, San Juan 00926-1119, Puerto Rico; mkeller.co2@gmail.com
4   NASA Jet Propulsion Laboratory, California Institute of Technology, Pasadena, CA 91109, USA; mlongo@jpl.nasa.gov
5   Embrapa Agricultural Informatics, Campinas, SP 13083-886, Brazil; maizanara@gmail.com
6   Neptune Inc., Lakewood, CO 80215, USA; pduffy@neptuneinc.org
7   NASA Goddard Space Flight Center, Greenbelt, MD 20771, USA; douglas.morton@nasa.gov
*   Correspondence: ekena.rangelpinage@student.uts.edu.au or ekenapinage@hotmail.com; Tel.: +1-(503)4732490

**Abstract:** Forest degradation is common in tropical landscapes, but estimates of the extent and duration of degradation impacts are highly uncertain. In particular, selective logging is a form of forest degradation that alters canopy structure and function, with persistent ecological impacts following forest harvest. In this study, we employed airborne laser scanning in 2012 and 2014 to estimate three-dimensional changes in the forest canopy and understory structure and aboveground biomass following reduced-impact selective logging in a site in Eastern Amazon. Also, we developed a binary classification model to distinguish intact versus logged forests. We found that canopy gap frequency was significantly higher in logged versus intact forests even after 8 years (the time span of our study). In contrast, the understory of logged areas could not be distinguished from the understory of intact forests after 6–7 years of logging activities. Measuring new gap formation between LiDAR acquisitions in 2012 and 2014, we showed rates 2 to 7 times higher in logged areas compared to intact forests. New gaps were spatially clumped with 76 to 89% of new gaps within 5 m of prior logging damage. The biomass dynamics in areas logged between the two LiDAR acquisitions was clearly detected with an average estimated loss of $-4.14 \pm 0.76$ MgC ha$^{-1}$ y$^{-1}$. In areas recovering from logging prior to the first acquisition, we estimated biomass gains close to zero. Together, our findings unravel the magnitude and duration of delayed impacts of selective logging in forest structural attributes, confirm the high potential of airborne LiDAR multitemporal data to characterize forest degradation in the tropics, and present a novel approach to forest classification using LiDAR data.

**Keywords:** tropical forests; forest degradation; selective logging; forest structure; laser scanning

## 1. Introduction

Forest degradation resulting from logging, fire, and other human causes is a widespread phenomenon in tropical regions that has significant impacts on the global carbon cycle [1–8]. Tropical forest degradation alters forest structure, species composition, and successional processes, causing impacts that may last for several years or even decades [9–12]. Degradation processes

are heterogeneous in space and time, especially when considering large areas such as the Amazon region [13]. Moreover, degraded forests are often located in dynamic frontiers where processes of forest regeneration and repeated disturbance increase the difficulty of classifying degraded forests and quantifying degradation effects [14,15]. The complex landscape dynamics in frontier forests contribute to high levels of uncertainty in the estimates of the extent of forest degradation and estimates of biomass stocks or changes associated with degraded forests [4,16,17].

Selective logging is one of the most common drivers of forest degradation and an important economic activity in tropical forest landscapes [2,5,18]. In this mode of timber harvest, only a few marketable trees are logged while, in principle, the remaining forest is left to regenerate until the next logging cycle. Logging modifies tropical forests' structural characteristics, forest composition, and ecosystem function [19–22]. Both conventional and reduced-impact logging (RIL) in tropical forests remove large canopy trees and damage residual neighboring trees, altering the forest micro-climate, live biomass, and species composition [23]. Increased mortality rates of damaged trees following logging have been documented with field studies [23,24].

Ground-based forest inventories contribute most to the current knowledge about selective logging but they are limited spatially and temporally because they are expensive. Remote sensing approaches have been investigated for both the classification of logged forest and estimation of the intensity of logging or logging damage. Passive optical remote sensing assessments at resolutions as fine as 20–30 m have been used to classify logged forests with moderate accuracy. However, they have struggled to quantify the extent, intensity, and duration of logging damage because of the lack of regular cloud-free repeat observations and because rapid canopy regeneration masks logging effects after as few as three years [25–27]. Moreover, a 30 m spatial resolution is not sufficient to resolve mortality of a single canopy tree [28]. Passive optical signals are dominated by upper canopy reflectance and provide limited information on the structure beneath top canopy layers [29].

Active remote sensing has great potential to overcome limitations inherent to forest inventory and passive optical remote sensing studies. More specifically, Light Detection and Ranging (LiDAR) remote sensing can be used to classify logged forests and to quantify changes in forest structure from selective logging over time. For instance, airborne LiDAR data is a valuable resource for fine-scale biomass and carbon stock estimation [16,30–32]. Over the past decade, several studies have used airborne LiDAR data across tropical regions to estimate the impacts of selective logging. For example, pioneering studies by d'Oliveira et al. [33] and Andersen et al. [34] identified logging damage using relative density models, estimated above-ground forest biomass, and quantified changes in forest structure with repeated LiDAR flights in the Western Amazon. In Sierra Leone, Kent et al. [10] distinguished structural differences between the canopies of logged and old-growth intact forests even 23 years after logging activities. In Indonesian Kalimantan, Ellis et al. [35] estimated the area affected by logging and biomass losses per management practice (hauling, skidding and felling of trees) for dipterocarp forests, while Wedeux and Coomes [36] assessed the canopy of old-growth and selectively logged peat swamp forests, and compared how canopy structural metrics varied with peat depth and logging status. Also working in Kalimantan, Melendy et al. [37] developed an automated approach for measuring the extent of selective logging damage that performed well, especially in the regular logging features created by mechanical means. That approach was later adopted by Pearson et al. [38] to estimate carbon losses for four forest concessions. Together, these studies demonstrate the great potential of LiDAR data from a single acquisition to accurately characterize and measure logging extent and impacts and post-logging gap and biomass dynamics in tropical forests.

In this paper, we extend the investigation of the effects of logging on forest structure using repeated LiDAR acquisitions at a site in Eastern Pará State in the Brazilian Amazon where RIL activities have been well documented since the 1990s. This site covers blocks logged with similar methods between 2006 and 2013. LiDAR data was acquired in both 2012 and 2014. We used the multitemporal data directly and also indirectly by trading space for time on the 8 year-long logging chronosequence for investigation of canopy and understory damage and recovery. We compared the rate of recovery for

canopy versus understory forest layers following selective logging. Additionally, we investigated whether canopy dynamics in the post-harvesting period are different from intact forest and how canopy damage is spatially and temporally related to previous selective logging damage. Using both the multitemporal data and the logging chronosequence, we tested a simple binary classification to separate logged and intact forest and demonstrated that logging can be accurately detected in the LiDAR data up to 8 years following the logging event. The combination of multitemporal LiDAR data applied to a logging chronosequence sets this work apart from previous studies.

## 2. Material and Methods

### 2.1. Study Area

Fazenda Cauaxi is a private property in the Paragominas Municipality of Pará State, Eastern Brazilian Amazon (48.48W, 3.75S). At this site, the Instituto Floresta Tropical (IFT) established a logging camp and school for demonstration of forest management techniques. Training courses, demonstration, and research activities have been conducted there since 1995 with the collaboration of property owners. Prior to current logging operations, there is no historical record of land use or collection of non-timber forest products although there are indications of indigenous activity [39–41].

The climate at Fazenda Cauaxi is humid tropical. Total annual precipitation averages about 2200 mm [42]. A dry season extends from July through November (generally <50 mm per month), although June and December are also often sufficiently dry for logging operations [40]. Soils in the area are mainly classified as dystrophic yellow latosols according to the Brazilian system [43]. The topography is mainly flat to mildly undulating [40] with the height above sea level in the study area ranging from 74 to 150 m (based on LiDAR-derived digital terrain model (DTM), discussed below). Forests at this site are classified as tropical moist, and *terra firme* (upland). They are diverse forests, with more than 124 tree species (diameter at breast height (DBH) $\geq$ 10 cm), emergent trees exceeding 50 m in height, and is characterized by a heavy vine load [39]. Forest inventory data estimate an average of 550 live stems ha$^{-1}$ (DBH $\geq$ 10 cm), and a mean aboveground biomass of ~179 MgC ha$^{-1}$ [16].

Our study area in Fazenda Cauaxi is divided into 12 harvesting blocks of approximately 100 ha each (2 intact and 10 logged from 2006 to 2013—Table 1, Figure 1). The nominal logging block boundaries and dates were provided by IFT based on their planning maps. Actual logging varied slightly from these plans, as discussed below. Logging was conducted using reduced impact techniques that included directional felling and careful planning of skid trails to limit collateral damage to non-harvest trees and canopy opening, and to minimize the area of ground disturbance [44]. Eighty timber species were harvested with a minimum cutting limit of diameter at breast height (DBH = 1.3 m) greater than 55 cm. The 5 most commonly harvested species were *Manilkara huberi, Manilkara paraensis, Chrysophyllum venezuelanense, Astronium lecointei,* and *Hymenaea courbaril,* accounting for nearly 50% of the total harvested volume.

The logging blocks were harvested with different intensities; areas logged in 2006, 2007, and 2008 had greater harvest volumes than those logged in 2010, 2012, and 2013 (Table 1), because the former areas are located on flat terrain (Figure 1). The influence of relief is often decisive in harvesting operations because optimal productivity and minimized understory damage are obtained on dry and flat terrain with an even surface and good load-bearing capacity [45]. To calculate the harvested volume per block, we summed the volume of all logs from cut trees (representing the trunks transported from the forest) of each block and normalized the sum to the available area for harvesting (defined according to a slope mask excluding pixels with slope greater than 20°, an operational criterion used by IFT). Harvest volumes are reported in m$^3$ ha$^{-1}$.

**Table 1.** Characteristics of harvesting blocks in Cauaxi site.

| Logging Block | Original Area (ha) | Year of logging | Harvested Volume (m³/ha) | Normalized Harvested Volume (m³/ha) † |
|---|---|---|---|---|
| B01 | 99.9 | 2010 | 11.03 | 12.05 |
| B02 | 104.9 | Intact | N/A * | N/A * |
| B03 | 99.9 | 2013 | 11.73 | 14.38 |
| B04 | 99.9 | 2010 | 21.46 | 23.79 |
| B05 | 104.9 | 2012 | 24.51 | 26.65 |
| B06 | 99.9 | 2012 | 14.99 | 19.32 |
| B07 | 99.9 | 2008 | 23.60 | 23.62 |
| B08 | 99.9 | 2007 | 14.41 | 17.76 |
| B09 | 105.9 | 2007 | 27.82 | 28.10 |
| B10 | 99.9 | 2006 | 21.29 | 21.29 |
| B11 | 99.9 | 2008 | 22.79 | 23.60 |
| B12 | 100.5 | Intact | N/A * | N/A * |

* N/A: not applicable. † Normalized block area removes all areas with slopes >20 degrees.

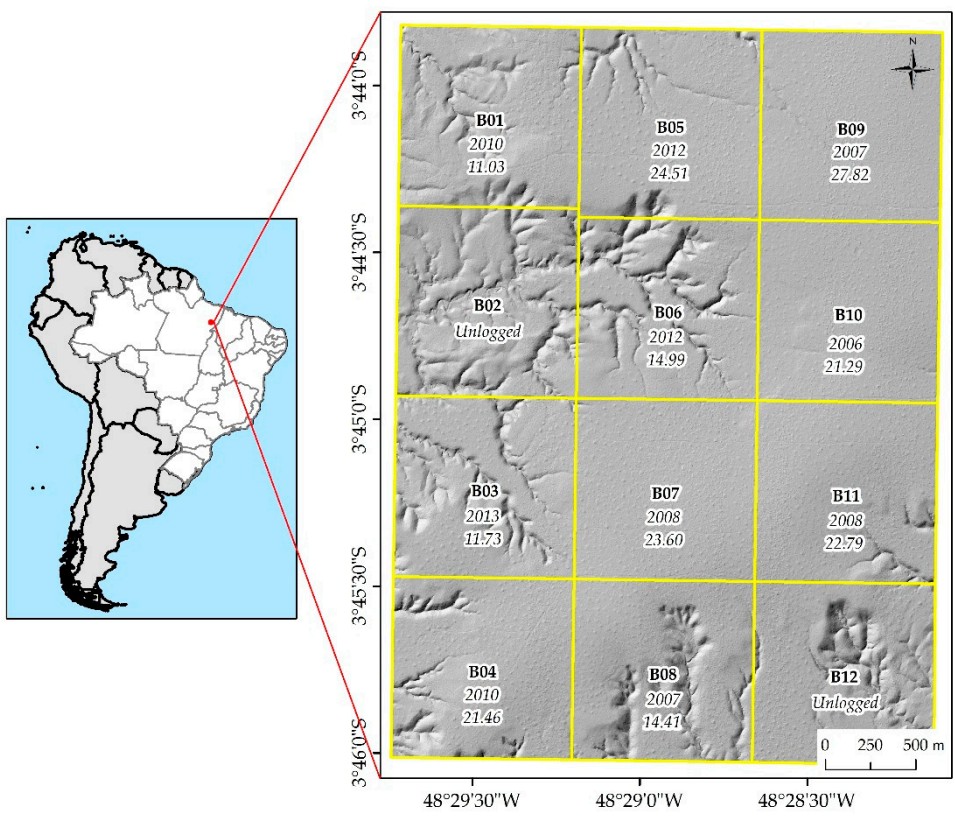

**Figure 1.** Location of the study area. The distribution of the blocks over the shaded relief of Cauaxi site (~1200 ha), year of logging, and harvested volume (m³ ha⁻¹) are included.

## 2.2. LiDAR Data and Processing

The LiDAR data were acquired on two dates (July 2012 and December 2014), yielding two high-density discrete return LiDAR datasets [46]. Both datasets were acquired by the same vendor, GEOID Laser Mapping (Belo Horizonte, Brazil), using similar LiDAR instruments from the same manufacturer, and with similar flight and data acquisition characteristics (Table 2). The vendor delivered the point cloud data with classified ground returns. The horizontal (x,y) and vertical (z) positional accuracies (1 sigma) of LiDAR returns were <42.5 cm and <15 cm, respectively based on the specifications of the product manual and the flight heights above ground. To ensure a sufficient

number of ground returns to characterize the terrain elevation, a minimum return density of 4 m$^{-2}$ over 99% of the surveyed area was required [47].

**Table 2.** Details of LiDAR data acquisitions.

| Characteristic | 1st Coverage | 2nd Coverage |
|---|---|---|
| Equipment | Optech ALTM 3100 | Optech ORION M300 |
| Acquisition date | 27–29 July 2012 | 26–27 December 2014 |
| Flight maximum height (from the ground) | 850 m | 850 m |
| Field of view | 11° | 12° |
| Laser pulse frequency | 100 kHz | 275 kHz |
| Fraction of area with return density >4 m$^{-2}$ | 99.35% | 99.99% |
| Mean return density | 28.3 m$^{-2}$ | 61.3 m$^{-2}$ |
| Mean first return density | 13.8 m$^{-2}$ | 37.5 m$^{-2}$ |
| Percentage of flight line sidelap | 65% | 65% |

Data were processed using established methods for NASA Goddard's LiDAR, Hyperspectral, and Thermal (G-LiHT) Airborne Imager [48]. In addition, we combined the ground returns from both surveys to generate a single DTM at 1-m horizontal resolution, in order to generate a more accurate LiDAR-derived DTM [49,50]. Accurate terrain characterization is required for accurate lidar-derived forest metrics [47,51], and the combination of two surveys improves the quality of the DTM and reduces uncertainty from subsequent multitemporal analysis. The terrain height derived from the combined DTM was subtracted from each return to remove the topographic influence on the forest height. A Canopy Height Model (CHM) was created by selecting the greatest return height in every 1 m grid cell, using these points to create a triangulated irregular network (TIN) and interpolating canopy heights on a 1 m raster grid [48]. A previous study in dense tropical forest using data from the same vendor with similar flight and instrumentation characteristics found that the vertical accuracy (as RMSE) of the 1 m resolution DTM was less than 1 m compared to a highly accurate ground based GNSS survey of 35 control locations [47].

Based on the CHM, we estimated canopy gap areas by defining a gap as any contiguous area with height <10 m and area >10 m$^2$ [50]. To assess the forest understory damage, we used a Relative Density Model (RDM) [33], calculated as the proportion of returns up to 0.5 m compared to the total returns, also at 1 m horizontal resolution. We classified polygons of understory damage as contiguous areas greater than 10 m$^2$ (same area threshold as the gap definition) containing more than 10% of returns below 0.5 m height. Polygons of both canopy and understory damage from the main road in blocks 01, 05 and 09 were excluded from the analysis.

We also assessed changes in gap area between 2012 and 2014 LiDAR collections. New gaps were considered as an estimate of canopy turnover and were computed as the non-gap area that became gap area over the sampling period based on the 10 m$^2$ area and 10 m height threshold [50]. In order to evaluate the effect of older logging activities on canopy turnover, we computed the minimum distance of each new 2014 gap polygon to understory damage polygons derived from the 2012 LiDAR data. We then calculated the proportion of gap polygons within distance ranges of 0 to 5 m, 5 to 10 m and >10 m of understory damage. We excluded the blocks logged between LiDAR collections (in 2012 and 2013) from this analysis, because it is not possible to separate new gaps caused by intentional felling from those caused by other causes (e.g., natural tree fall) in these blocks.

To investigate biomass dynamics between the LiDAR collections, we used the LiDAR-biomass model from Longo et al. [16], at 50 m spatial resolution. This study [16] quantified the spatial distribution of above-ground carbon density (ACD) on intact and degraded forests by combining a network of 359 forest inventory plots (86 ha) with 18,000 ha of airborne LiDAR surveys, including the Cauaxi data from 2012. Longo et al. [16] presented two calibrations linking ACD and LiDAR metrics, one that relied on multiple metrics and a second that relied solely on top of canopy height (TCH). Both models had similar uncertainties, so we selected TCH because it is likely to be more robust to

differences in flight and LiDAR acquisition characteristics [52]. We generated an ACD difference map by subtracting the 2014 estimate from the 2012 one, and used the difference map to assess biomass changes in the blocks harvested in different years and in intact forests.

## 2.3. Data Analysis

We tested whether logged forests show greater amount of canopy gaps, understory damage, new gaps and biomass changes compared to intact areas. For this purpose, we randomly allocated 100 square samples of 0.25 ha (50 × 50 m) over each treatment (intact, logged in 2006, logged in 2007, logged in 2008, logged in 2010, logged in 2012, and logged in 2013) and extracted the area of each of these variables for each 0.25 ha plot. Plots were allocated outside of the slope mask because we assumed that logging activities were not conducted in steep slopes.

Next, we tested the normality of these variables' distributions using the Shapiro-Wilk test, and found that none of them had normal distributions (all *p*-values <0.01). Therefore, we carried out a two-sample Wilcoxon signed rank pairwise non-parametric test, adjusted using the Bonferroni correction, to compare each of the 21 possible pairs of groups of logging treatments.

To classify logged versus intact areas, we developed a general approach based on an *a priori* classification using the 2012 LiDAR data coverage. Because the removal of large trees by logging activities leads to a change in LiDAR vertical profiles (frequency of LiDAR returns versus height), we examined a variety of LiDAR metrics and developed a statistical model to classify intact versus logged status using information from the third and fourth moments (i.e., skewness and kurtosis) of the distribution of return frequency versus height. Using skewness and kurtosis is relevant because they represent a measure of the degree of within-pixel symmetry and peakedness/flatness in the point cloud density distribution, respectively. These statistics are good descriptors of selective logging disturbance: the removal of large trees by loggers usually causes a general shift in canopy height, with a decrease of tall canopy returns and a relative increase of short canopy return frequency due to the canopy gaps created. We used generalized boosted models to implement our classification approach due to their ability to depict complex nonlinear relationships with a minimum of assumptions that need to be met [53].

The classification model was implemented with the gbm package in R with the *adaboost* distribution option, in order to estimate the logged/intact probability for each pixel. Each pixel in the 2012 domain of interest was classified *a priori* as either intact or reduced impact logging (RIL) using spatially explicit historical information provided by IFT. For pixels that intersected the boundary of the domain of historical logging activity, we selected the class that covered the majority of the pixel area. This binary classification (i.e., intact or RIL) was used as the response variable in the statistical model development. Values of the explanatory variables for each pixel were derived from the skewness and kurtosis of the within pixel distribution of the LiDAR data.

A randomly selected 90% subset of the pixels was used to build the model and the classification accuracy was assessed using Area Under the Curve (AUC) criteria on the remaining 10%. The AUC is a common approach used to assess the performance of a binary classifier [54]. The cv-folds method (25 fold cross-validation) was used to estimate the optimal number of decision trees for the classification models. This approach was used for prediction at the 10% subset of pixels that were not used to build the model. The bag fraction (fraction of the training set observations randomly selected to propose the next tree in the expansion) in the gbm model was set to 0.6, resulting in a stochastic model fitting approach. To generate a measure of uncertainty associated with the model fit that accounts for both sample variability (with respect to the partitioning of the data to test and train subsets) and model stochasticity associated with bag fraction less than 1, we fit 100 different models, each corresponding to different subsets of the data. For each randomly selected subset, a new gbm model was fit and AUC was used to assess fit. Figure 2 graphically summarizes the workflow of the analysis we performed.

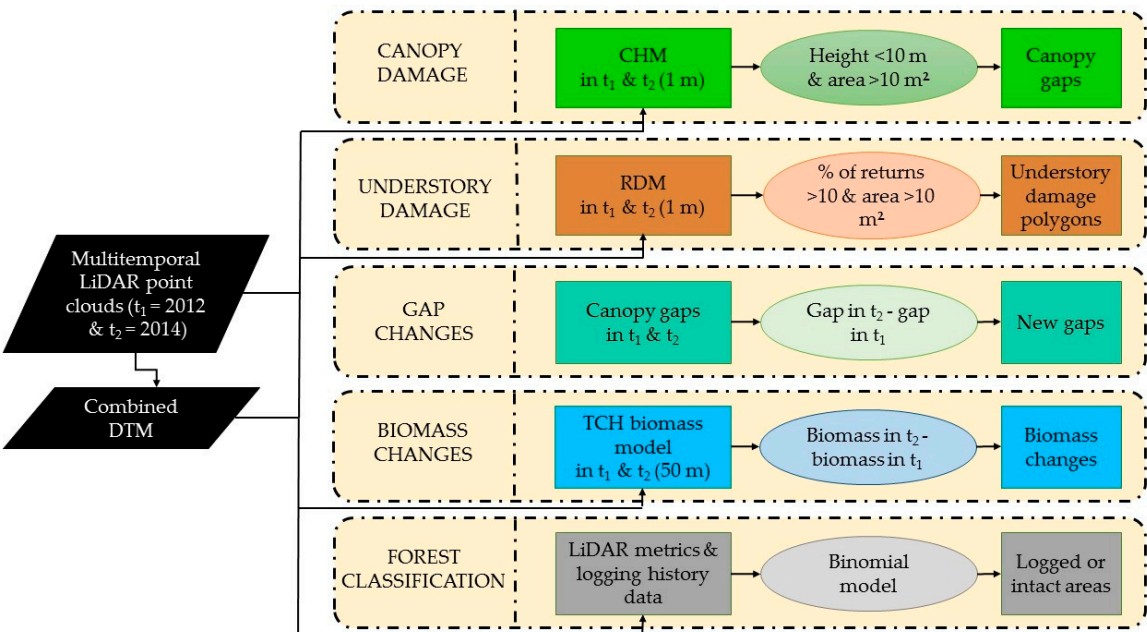

**Figure 2.** Workflow summaries for four major variables and the forest classification.

## 3. Results

### 3.1. Canopy Versus Understory Damage

Both recent and earlier changes in forest height due to logging can be observed in the canopy height models for 2012 and 2014 (Figure 3A,B). The primary access road (in B01, B05, and B09) that crosses the area is also visible through the canopy (Figure 3A,B). Most of the large (up to 20 m) decreases in canopy height are found in the areas that were harvested between 2012 and 2014 (B03, B05, B06; Figure 3C), although small clusters of canopy height losses from canopy tree fall and branch fall events occurred in all blocks, regardless of logging history.

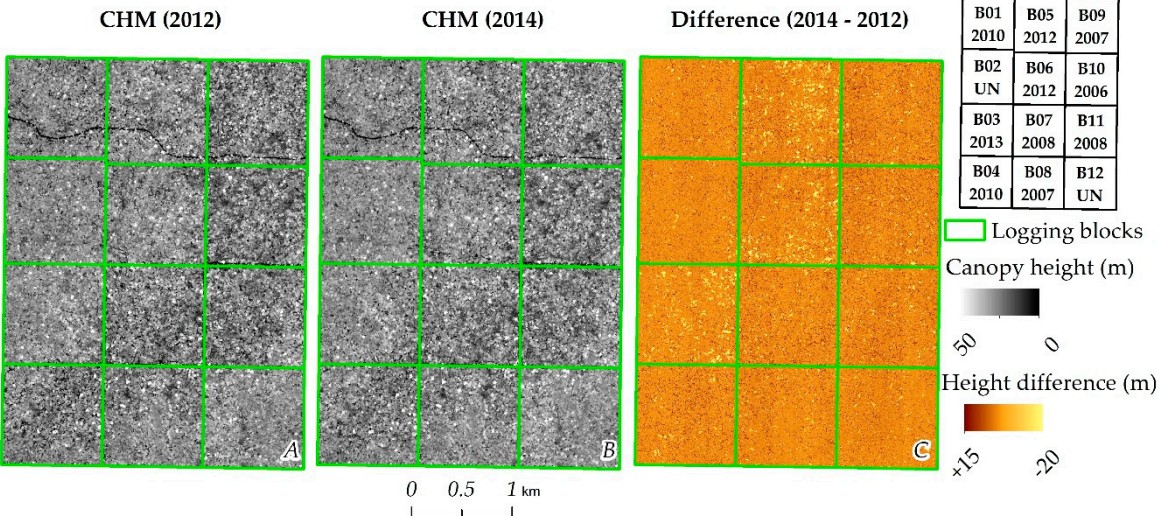

**Figure 3.** Canopy Height Models for 2012 (**A**) and 2014 (**B**) LiDAR coverages, and difference image (**C**) for Cauaxi site. The code and year of logging for each logging block are shown in the index map in the top right (UN stands for unlogged).

The understory damage maps clearly show the absence of shorter vegetation along secondary roads and skid trails, in addition to the primary road (Figure 4). Some logging infrastructure has characteristic shapes; secondary roads are often designed with loops for trucks to maneuver (blue arrows in Figure 4D,E), whereas decks for log storage are larger polygons along the roads where all vegetation has been cleared (purple arrows in Figure 4D,E). Understory damage is abundant throughout recently logged areas (Figure 4A) and gradually disappears over time (Figure 4B,C), especially for areas logged between 2006 and 2008. The greatest positive differences in return proportion (light yellow in the difference image, Figure 4C) indicate new areas of understory damage in blocks B03, B05, and B06, consistent with canopy damage from logging between LiDAR collections in these blocks (Figure 3). Negative differences between the 2012 and 2014 understory damage estimates (green in the difference image, Figure 4C) indicate the recovery of understory vegetation in older logged blocks (e.g., B01 and B04).

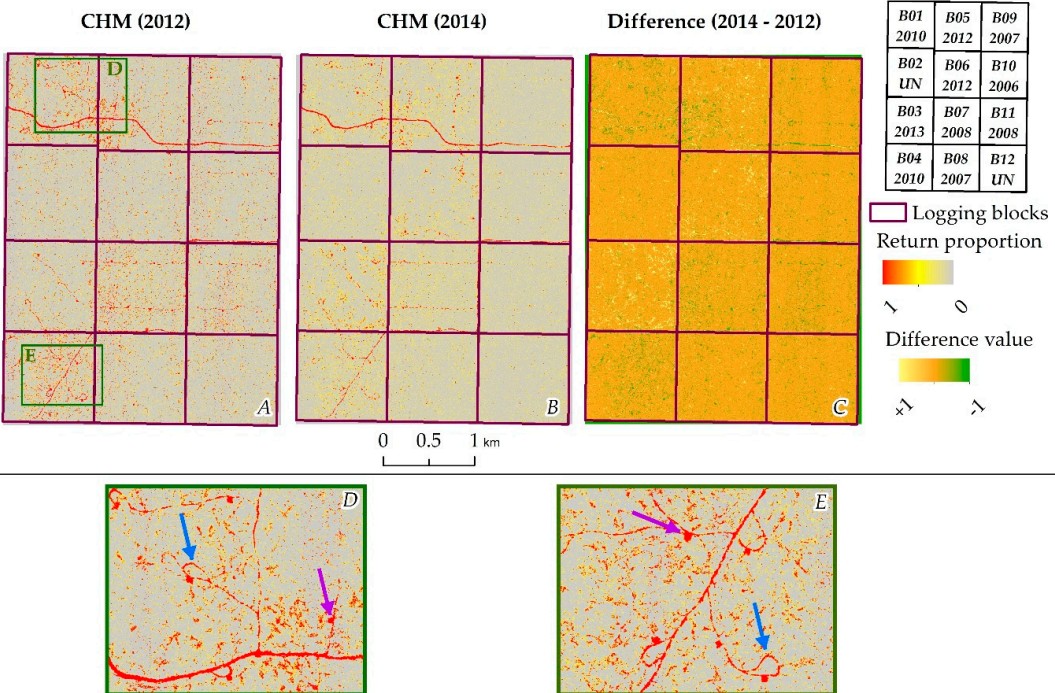

**Figure 4.** Understory damage maps, derived from LiDAR RDM for 2012 (**A**), 2014 (**B**), and the difference between 2012-2014 (**C**) for Cauaxi site. The code and year of logging for each logging block are shown in the index map in the top right. Zoomed insets (**D**,**E**) show understory damage caused by roads and road loops (indicated by blue arrows) and log decks (indicated by purple arrows).

Intact areas had less canopy and understory disturbance compared to logged areas. Considering the total area available for each logging treatment (i.e., excluding the slope mask area), we found an average of 2.82 and 5.19–7.70 (average per year of logging) canopy gaps per ha$^{-1}$ in intact and logged forests, respectively. In terms of area, we found 109 and 205–437 m$^2$ of canopy gaps per hectare in intact and logged forests, respectively (Figure 5A,B, Figure 6A). Likewise, we found a larger number of understory damage polygons in logged blocks (5.61–16.57 per ha$^{-1}$) than in the intact forests (7.71 per ha$^{-1}$); and a larger area as well (182–1558 and 227 m$^2$ ha$^{-1}$ for logged and intact forests, respectively, Figure 5C,D, Figure 6B).

The signal of elevated canopy damage in logged forests persisted for at least 8 years after wood harvest while understory damage was compensated for by regrowth during this period. The ground signal was much stronger in the areas recently logged, fading in the older areas (242 m$^2$ ha$^{-1}$ for areas logged in 2006, and 1558 m$^2$ ha$^{-1}$ for areas logged in 2013, Figure 5C,D, Figure 6B), whereas the

canopy damage signal was much more homogeneous across logged areas, regardless of year of the logging (258–391 m$^2$ ha$^{-1}$ for areas logged in 2006 and 2013, respectively, in Figure 5A,B, Figure 6A).

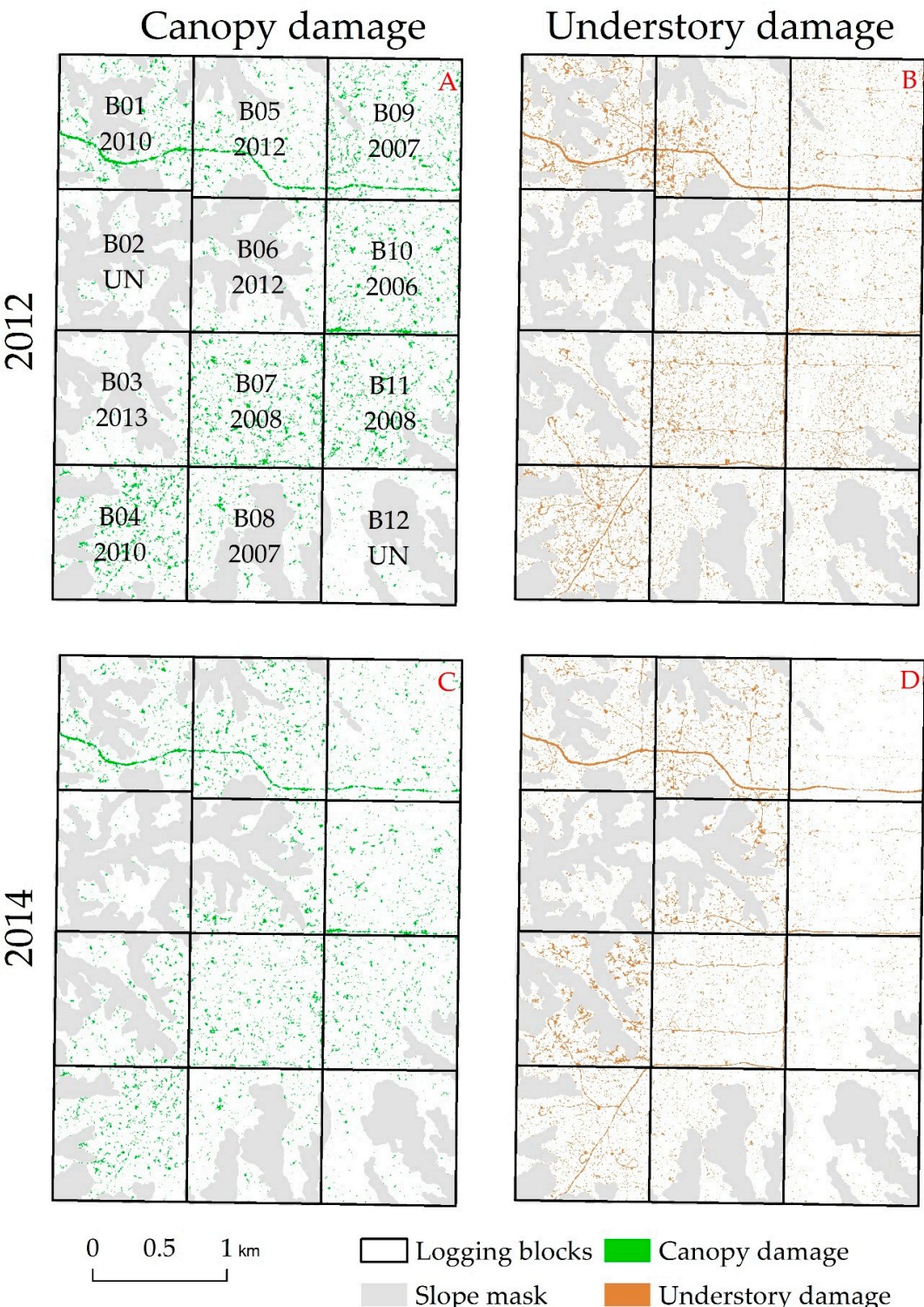

**Figure 5.** Canopy (panels **A** and **C**) and understory (panels **B** and **D**) damage for 2012 and 2014 LiDAR coverages in Cauaxi site. Block code and year of logging are shown in panel **A**.

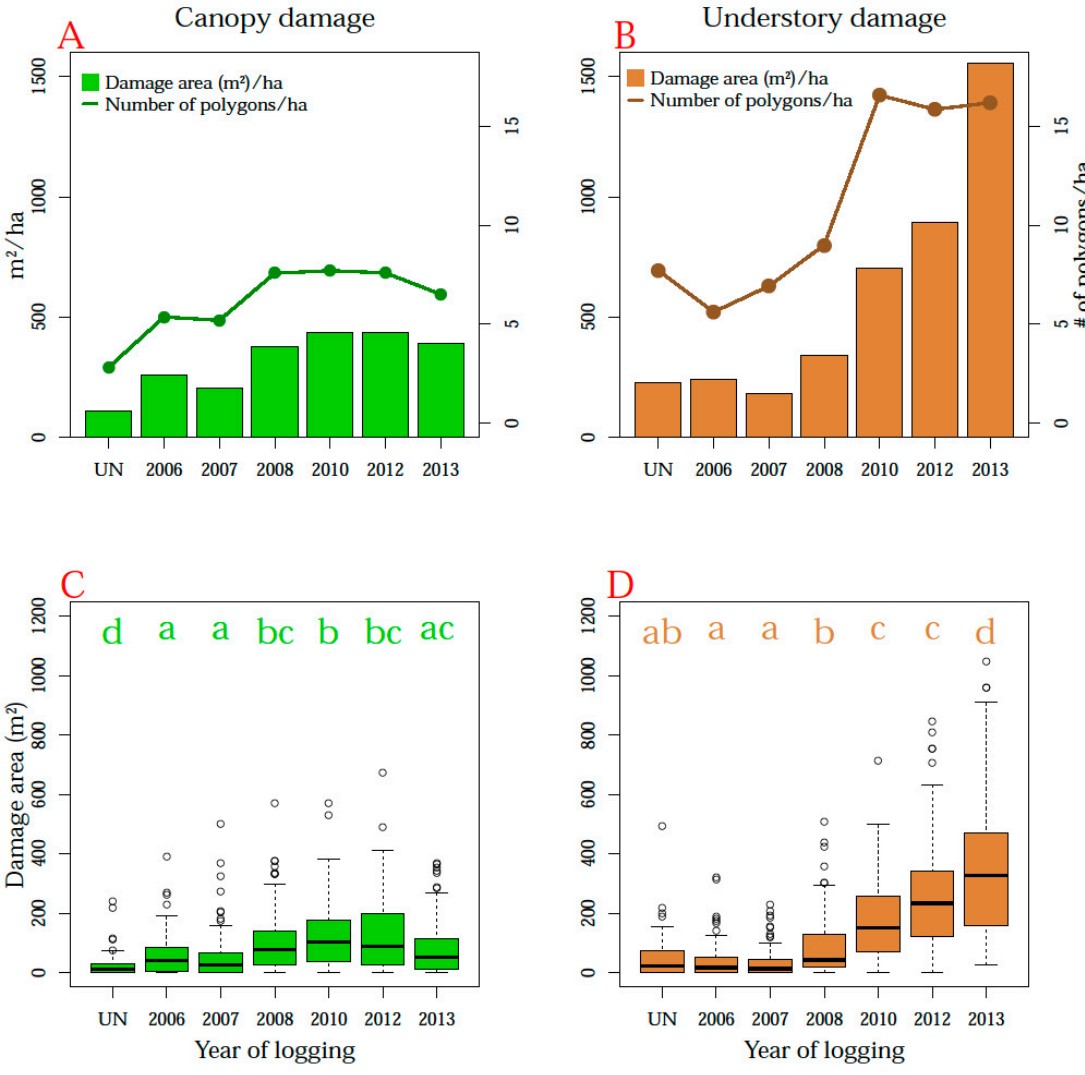

**Figure 6.** Top panels (**A**,**B**): canopy and understory damage per year of logging for 2014 LiDAR coverage. Bars show the total damaged area per hectare and lines represent the number of damage polygons per hectare (considering total available for each category, slope mask area excluded from estimates). Bottom panels (**C**,**D**): box-and-whisker plots of canopy and understory damage from the 0.25 ha samples at the logging treatments. Groups labelled with the same letter are not significantly different at confidence level of 99% (Wilcoxon test). Box heights correspond to the interquartile range, whiskers span up to 1.5 times the interquartile range, and the horizontal line corresponds to the median.

Logging activities in Cauaxi showed multi-year damage to both the canopy and understory, although important differences exist between the types of damage. For canopy gaps, there was strong evidence ($p$-value $< 0.01$) of significant differences among intact areas and all the other groups (Figure 6A,C). The mean area ($\pm$standard error) of canopy gaps in the intact forest samples (n = 100 for each logging treatment, each sample plot = 0.25 ha or 2500 m$^2$) was $23 \pm 3.8$ m$^2$, while the mean in the logged treatments ranged from $54 \pm 8.2$ to $123 \pm 12.3$ m$^2$. Understory damage, on the other hand, decreases with time since logging (Figure 5B,D), reflecting the ongoing understory recovery. The ground damage signal from intact forest samples was not significantly different from that originating in areas logged in 2006, 2007, and 2008 ($p$-value $< 0.01$). The mean area of understory damage polygons in the intact forest samples was $47 \pm 6.7$ m$^2$, while the mean in the logged treatments ranged from $33 \pm 5.0$ to $354 \pm 24.0$ m$^2$. This pattern shows that understory recovery in these forests occurred within 6

to 7 years after harvesting. In contrast, canopy recovery was incomplete within the 8-year span of our study (Figure 6C,D).

New gap formation, an estimate of canopy turnover, was also lower in unlogged forests than in unlogged stands. Considering the total area available, we found an average of 43 m² ha$^{-1}$ of new gaps within intact forests, and 76–290 m² ha$^{-1}$ within logged forests. The smallest rate of new gap formation, considering the full observation period of 2.5 years, was found in the intact plots (0.42%), while the greatest rates were found in the plots logged between LiDAR acquisitions (2.89% in 2012 and 2.39 % in 2013). The blocks logged before LiDAR acquisitions showed intermediate and variable rates (0.76 to 1.19%—Figure 7). Harvesting operations in Cauaxi were associated with a gap formation rate 5.6 to 6.8 times greater than rates for intact forests. In the areas logged prior to 2012, the gap dynamics between 2012 and 2014 are influenced by natural and post-harvesting effects and not by intentional felling of trees by loggers. These pre-2012 logged areas present greater gap formation rates than intact forests (1.8 to 2.8 times), but with an inconsistent pattern related to year since logging (Figure 7A). New gap formation was significantly higher (*p*-value < 0.01) in all logged samples than in the intact areas' samples, except for areas logged in 2010 (Figure 7B). The mean (±standard error) area of new gap polygons (including those with zero gap area) in the intact forest samples was 9 ± 2.5 m², while the mean in the logged treatments ranged from 24 ± 5.1 to 88 ± 11.2 m².

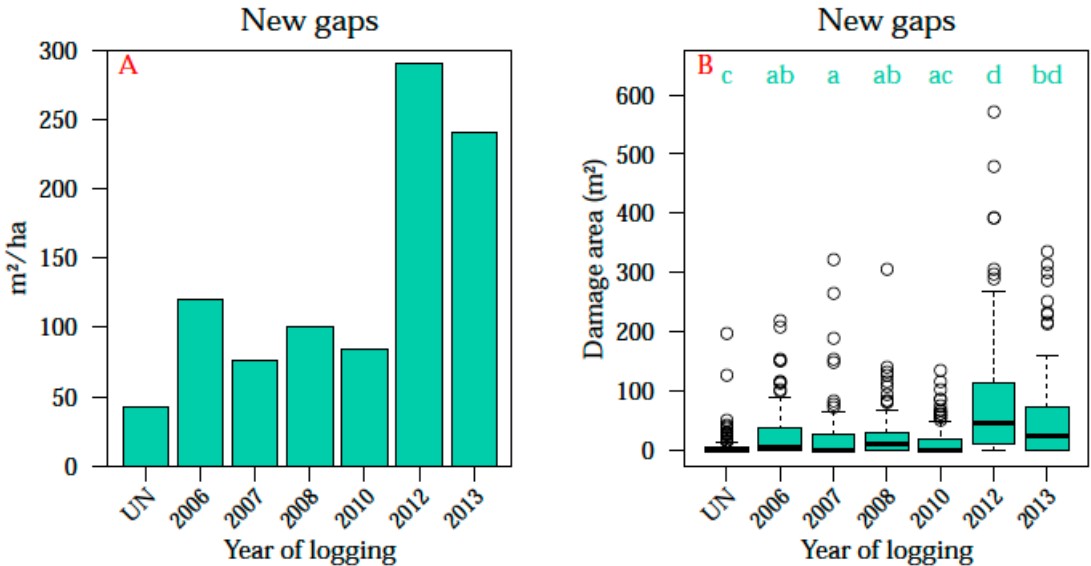

**Figure 7.** Area with new gaps between the two LiDAR acquisitions at the logging treatments (**A**) and distribution of new gaps from the 0.25 ha samples (**B**). In panel B, groups labelled with the same letter are not significantly different at the 99% confidence level (Wilcoxon test).

New canopy gaps were found to be highly clumped around logging-impacted areas (Figure 8). Using the understory damage mapped in 2012 as an indicator of logging impact, we found that 76% of new gaps that formed in areas logged in 2006 and 89% of new gaps that formed in areas logged in 2010 were within 5 m of previous logging damage. Furthermore, only 14% (2006) and 4% (2010) of new gap polygons were further than 10 m from understory damage areas (Figure 8). For this analysis, we considered the understory damage of the areas logged prior to the first LiDAR acquisition only.

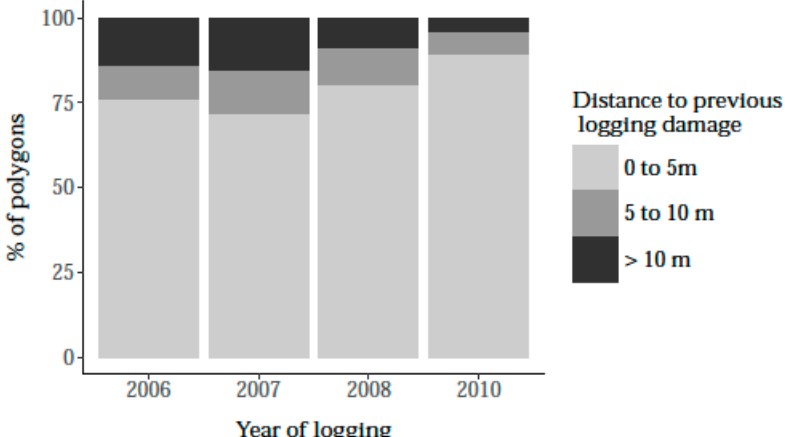

**Figure 8.** Percentage of new gap polygons (masked out by slope) within each distance group from the 2012 understory damage, by logging year.

*3.2. Biomass Changes*

LiDAR-based biomass estimates were shown to be responsive to logging-induced changes. The above-ground carbon density (ACD) maps (Figure 9) clearly depict the impacts concentrated on the flat plateaus in our study area, and show that the TCH LiDAR calibration is highly sensitive to ACD changes due to canopy tree removals by logging activities. Areas logged between 2012 and 2014 show concentrated losses of biomass (B03, B05, and B06, Figure 9), whereas older logging blocks and unlogged areas show heterogeneous biomass losses and gains, consistent with areas of canopy turnover from tree and branch fall (Figure 5).

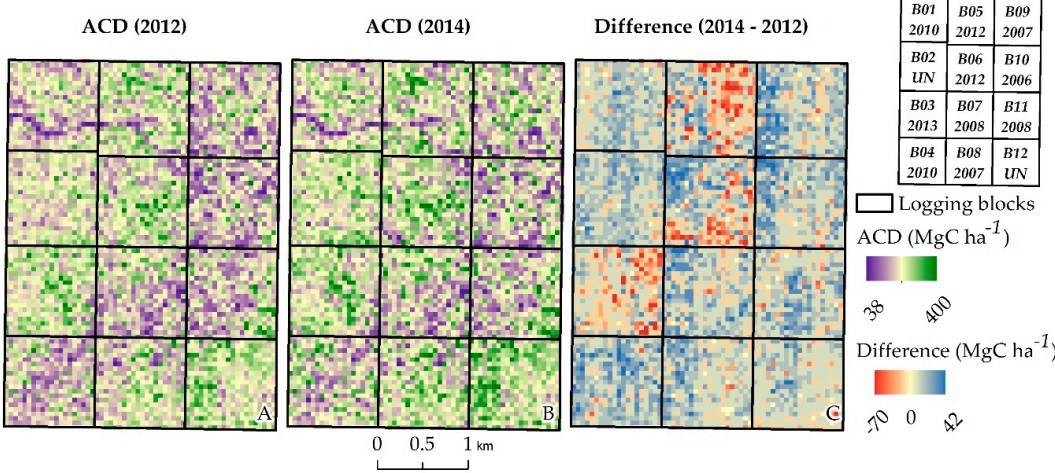

**Figure 9.** Above-ground carbon density for 2012 (**A**) and 2014 (**B**) and difference image (**C**) in Cauaxi site. The code and year of logging for each logging block are shown in the index map in the top right.

Intact forest areas and areas logged between 2006 and 2010 presented slight biomass gains between 2012 and 2014. In the 100 selected samples for each logging treatment, we found $8.18 \pm 0.93$ (mean $\pm$ standard error) MgC ha$^{-1}$ for intact and $10.53 \pm 0.52$ MgC ha$^{-1}$ for logged forests (Figure 10). In contrast, recently logged areas (2012–2013) lost carbon ($-10.35 \pm 1.90$ MgC ha$^{-1}$), albeit with a higher range of variation. When we compared recently logged areas to older logging and intact forests, we found significant differences ($p < 0.01$; Wilcoxon Rank Sum test) in ACD (Figure 10). ACD changes in intact forest plots were not significantly different from the changes in areas logged in 2006, 2007, and 2008.

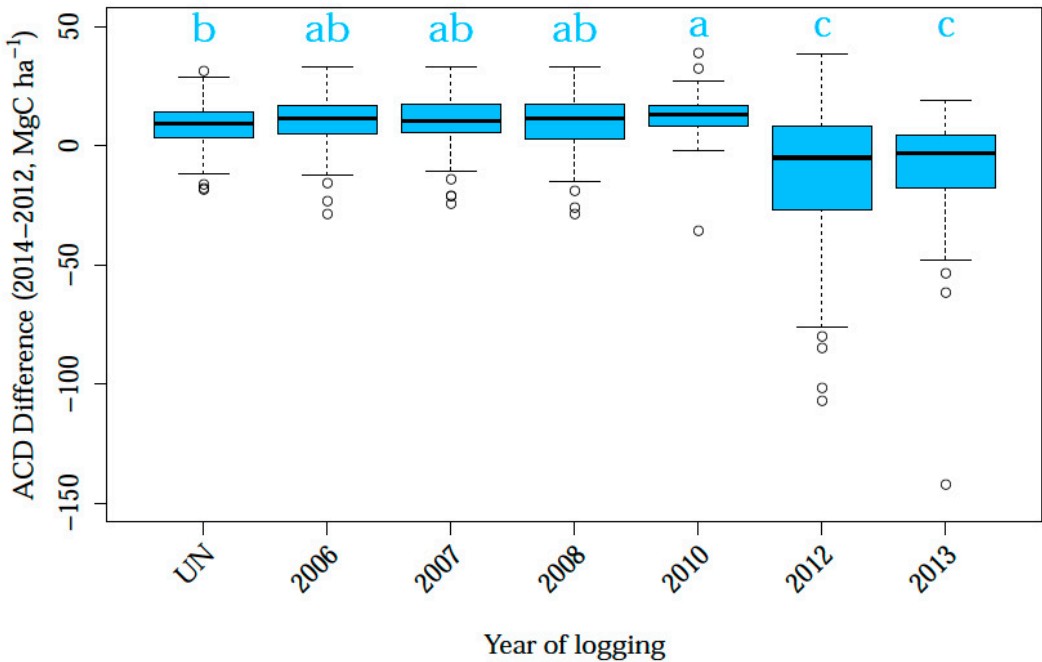

**Figure 10.** Distribution of ACD differences in the 0.25 ha samples at the logging treatments. Groups labelled with the same letter are not significantly different at 99%confidence level (Wilcoxon test).

### 3.3. Classification of Intact Versus Degraded Forest

We used the IFT records of logging activities to calibrate a classification model of logged versus intact forest status, based on 2012 LiDAR data for spatial resolutions from 50 to 250 m (Table 3). The classification model performed well across all spatial resolutions, as measured by the AUC (that ranged from 0.76 to 0.86; Table 3, second column). When moving from fine to coarse spatial resolutions, there was an increase in model performance; however this effect was most pronounced (i.e., largest difference in AUC when moving from fine to coarser resolution) when moving from 50 m to 100 m resolution (Table 3). There was also a trend of increasing SD in the AUC model fit estimate with an increase in the spatial resolution. To some extent, this is a function of the relative difference in the number of pixels as a function of spatial resolution. With a randomly selected 10% of the data held out for each iteration of the model building, with the 250 m resolution data this corresponds to 23 pixels, yet with the 50 m resolution, it is over 500 pixels.

**Table 3.** Model fit for varying resolutions. SD represents the standard deviation of AUC values when models were repeatedly fit using 100 randomly selected subsets of the data (2012) and the entire LiDAR dataset (2014).

| Resolution (m) | AUC-Mean (2012) | AUC-SD (2012) | AUC Mean (2014) | AUC-SD (2014) |
|---|---|---|---|---|
| 50 | 0.76 | 0.03 | 0.79 | 0.001 |
| 100 | 0.82 | 0.05 | 0.86 | 0.002 |
| 166 | 0.85 | 0.09 | 0.91 | 0.003 |
| 250 | 0.86 | 0.10 | 0.90 | 0.004 |

The total uncertainty associated with the AUC corresponding to the models built with the 2012 LiDAR data has two components: the impact of sample variability and stochasticity of the gbm model fit (relevant when bag fractions are less than 1). To understand the relative contributions of each component, we computed the AUC for the same model fit applied to the 2014 LiDAR coverage without any additional training. AUC values also increased as a function of resolution. The SD of the AUC from the application to the entire 2014 dataset is about 30 times smaller than that computed on the 10%

test subsets from the 2012 dataset. This suggests that the impact of model stochasticity is negligible when compared to the impacts of sample variability. It also indicates that the classification approach is robust with respect to several years' gaps between model fit.

## 4. Discussion

In this study, we used LiDAR and logging history information to assess how logging affected the dynamics of the understory, canopy, and biomass of a tropical forest in Eastern Amazon. Using the same information, we developed a probabilistic classification to distinguish logged and unlogged forest. Our study depended upon a unique combination of repeated LiDAR coverages and 8 years of logging records that allowed us to track forest disturbance and recovery across a chronosequence of different aged sites. The chronosequence, also called space-for-time substitution, assumes that spatial and temporal variation are equivalent. Although this assumption has some limitations, studies relying on space-for-time substitution are convenient and provide comparable datasets and a replacement for long-term studies [55]. Our study is an excellent case of space-for-time substitution because the forest areas selected for demonstration by the IFT foresters are homogeneous, and the logging practices and species harvested were tightly controlled.

We did not conduct an explicit ground-based validation during this study. For canopy gaps, we doubt that ground-based studies could measure canopy effects as accurately as LiDAR. For understory damage and biomass, ground-based validation may be useful, but it is costly. In the case of logging classification, we calibrated the model for one sampling epoch and tested the classification for a separate epoch on the same site. Undoubtedly, it would be useful to test the model at other sites. We plan to validate the classification approach in future studies.

### 4.1. Forest Damage from Logging

Our analysis clearly demonstrates that even reduced-impact logging resulted in approximately two to seven times more canopy damage compared to natural canopy turnover in adjacent unlogged forests (Figures 4 and 5). Canopy damage also exceeded intact forest rates in unlogged forests for at least 6 years following logging, suggesting that logging impacts led to delayed mortality over a long period. In contrast to canopy gaps, the understory vegetation disturbance was mostly erased by vigorous regrowth after 4–5 years. This is consistent with a field study by Schulze and Zweede [23] at the same site, who found that after 5 years, the upper canopy of logged stands was still fragmented while skid trails and gaps created by treefalls had dense regeneration of 3–8 m height.

Regeneration in the post-harvesting period is faster in the forest understory than in the canopy, and in some cases, there is less apparent understory damage in logged areas (e.g., in 2007, although the differences are not statistically significant; Figure 6) than in intact areas. This is in agreement with studies of natural forest gaps that have highlighted the relevance of increased light availability in gaps which favors fast-growing and light-demanding species [56–58]. Similarly, studies using airborne LiDAR demonstrated that vertical infilling (growth upward from the ground) has a much larger contribution to canopy change than horizontal infilling (when canopy position is lost to a neighboring contender tree) in tropical forests [50,59]. Furthermore, the delayed mortality related to logging activities, evidenced in our study (Figure 8), may also impair canopy recovery.

Both canopy and understory damage in logged forest differ from observations in intact forests, indicating that they are sensitive indicators of degradation immediately after logging. However, understory damage lasted no longer than five years in our study site, whereas we can say that canopy gaps are still detectable for at least eight years after logging. A previous study from Kent et al. [10] found long-term canopy effects caused by selective logging after 23 years in tropical African forests. Both understory damage and canopy damage have been studied individually [23,40,60], but to our knowledge, this is the first time that persistence of both signals has been compared in the same site. This allowed us to determine that the time scale of forest recovery from logging is vertically structured,

with the understory recovering faster than the upper canopy. The dynamics of the vertical distribution of leaf area across canopy microenvironments have a significant impact on forest function [61].

### 4.2. Forest and Biomass Dynamics

The combination of two LiDAR collections at Cauaxi along with the chronosequence of logged blocks allowed us to analyze canopy dynamics comparing intact forests, older logged forests, and forests logged between LiDAR acquisitions. We found that the rate of new gap formation is greater in logged areas than it is in intact forests, regardless of year of disturbance from 8 y to <1 y. Our estimates of canopy turnover in the form of new gaps are conservative because we did not account for canopy turnover events smaller than the minimum gap size adopted in this study (10 m$^2$) resulting from whole-tree mortality or partial mortality through branch-fall [62,63].

We also found that logging has a long-lasting impact on tree mortality because canopy losses continue to occur at increased rates for at least 8 years following the logging events. The rate of new gap formation is pronounced even in old logged areas compared to intact forests (0.30 to 0.48 y$^{-1}$ and 0.17% y$^{-1}$ respectively; Figure 7). Moreover, we observed that canopy turnover events are spatially clustered around the sub-canopy logging impact suggesting that post-logging mechanical effects such as bark scrapes, soil compression, and associated root damage may be linked to subsequent mortality. Leaning trees and canopy damage caused by treefalls also contribute to the delayed mortality that tends to decrease in the years following logging (Figure 8). Partial crown damage has been shown to predict mortality in intact forest in the Lambir Hills National Park in Malaysia [64]. Thus, estimates of carbon emissions caused by logging will be underestimated if they only account for immediate mortality of trees [65].

Our findings of delayed mortality following logging are consistent with field studies in other tropical forests. Pinard and Putz [66] found for dipterocarp forests in Malaysia that 8 to 10% of trees with "other damage" (neither snapped nor uprooted) resulting from logging subsequently died 8 to 12 months after timber harvest. Working in Bolivia, Pearson et al. [65] reported that 28% of trees recorded as leaning after timber harvest were dead four years after initial logging. In French Guiana, mortality rates and basal area loss (less skewed towards small trees than mortality) remained elevated for 3–8 years following logging, depending on logging treatment [9,67]. Shenkin et al. [68] found that mortality rates of trees damaged by RIL in Amazonian Bolivia peaked in the first year after logging, and then gradually declined to background rates in 8 years. These studies point to the need for investigations with larger time spans and different logging intensities to determine the persistence of logging impacts on the canopy.

Using repeated LiDAR acquisitions, we estimated substantial ACD changes caused by logging activities (mean (±standard error) biomass loss of −12.4 ± 3.16 and −8.3 ± 2.14 MgC ha$^{-1}$ for 2012 and 2013 harvests, respectively (Figure 9)). The large uncertainties arise from the uneven distribution of damage, including the large gaps caused by the felling and collateral damage. We estimated slight ACD gains in the other area and slightly larger gains in forests logged prior to the first LiDAR acquisition than in intact forests (mean biomass gains of 8.2 ± 0.94 MgC ha$^{-1}$ for intact forests, and a range of 9.2 ± 1.12 to 12.7 ± 0.84 MgC ha$^{-1}$ in areas logged from 2006 to 2010). Interestingly, logged forest and intact forest appear to be gaining carbon at similar rates, perhaps reflecting the balance between increased growth and increased mortality at the logged sites (Figure 10). In contrast, Figueira et al. [69] found overall ACD declines and elevated mortality for at least three years after logging in the Brazilian Amazon, and Blanc et al. [9] demonstrated that logged plots remained sources of carbon emissions for 10–12 years following harvesting activities at Paracou (French Guiana).

### 4.3. Scale Effects on the Logging Classification

Estimation of tropical forest degradation area and damage using remote sensing products present huge discrepancies due to differences in definitions, methodologies, and orbital sensors employed [70]. As a step toward improving the accurate identification of degraded forest, we generated a binary

classification statistical model to predict the probability of reduced impact selective logging in tropical forest from 0 to 8 years following timber harvest. Compared to conventional logging, changes in the forest due to RIL logging are minimal. Therefore, if our approach accurately classifies RIL logging, it should easily detect the far more common and damaging conventional logging.

Classification performance was scale-dependent, with model fit improving up to the largest grid size tested, 250 m. Scale dependence arises from a combination of both the dimensions of logging impacts and the use of higher order moments as explanatory variables in the classification models. The area of the logging blocks is 16 times larger than the coarsest resolution considered. Therefore, the spatial heterogeneity associated with the data used to delineate intact versus RIL areas is fairly low. At smaller spatial resolutions, the perimeter versus area ratio increases and edge effects (e.g., leaning trees, imprecise historical boundaries) are more likely to obfuscate information contained in the *a priori* classification. For example, it is possible to identify areas within the blocks designated as intact, that appear to have experienced some degree of logging. The NE corner of B12 is a region where this effect appears (Figure 11). This leads to larger uncertainty in the analysis using the smaller resolutions, both due to edge effects and the number of pixels that could be incorrectly classified.

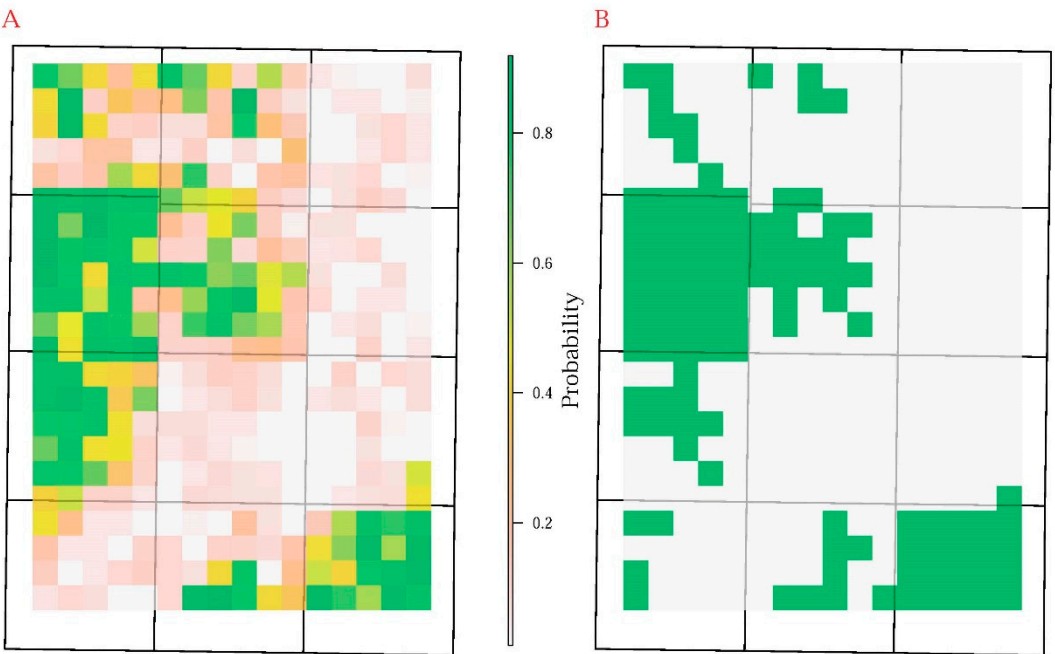

**Figure 11.** Classification (at 166 m resolution) giving the probability that the forest is intact based on 2012 LiDAR data (**A**). Intact forest (green) and logged forest (grey) at the Fazenda Cauaxi based on logging records by IFT (**B**).

The use of higher order moments shifts the focus to the tails of the within-pixel distribution of the CHM. Larger pixels have a greater number of LiDAR returns that allows for better characterization of the higher order moments, providing a clearer signal for the statistical model. Where there is sub-pixel heterogeneity at the larger resolutions (e.g., the NE corner of B12), the higher order moments are relatively robust to small pockets of logged areas in pixels classified as intact and vice-versa. The use of higher order moments is effective at identifying pixels where some RIL has occurred, but sub-pixel heterogeneity exists. Our model results suggest that for the spatial pattern of RIL disturbance in this region, the 166 m resolution analysis is optimal.

One advantage to this classification approach is the probabilistic nature of the statistical model output, which provides an estimate of the probability of intact versus logged status for each pixel. Classification uncertainties can be incorporated with those from other variables (e.g., biomass) and model building steps to estimate forest degradation effects and corresponding uncertainties over

regions of interest. Large scale airborne LiDAR surveys are becoming available in tropical forest regions. For example, recent regional to national level LiDAR campaigns have been carried out in Democratic Republic of Congo [71] and Gabon [72,73]. Potentially, these data can be used with training data and a classification approach similar to ours to develop degradation status maps based on vegetation structure. Importantly, because we used moments of the vertical distribution of returns over areas with improved results at coarser resolutions, our findings suggest that we may be able to classify degraded forests with LiDAR products with larger coverages and footprints, such as the recently launched Global Ecosystem Dynamics Investigation (GEDI) instrument deployed on the International Space Station.

## 5. Conclusions

Forest degradation by selective logging is pervasive throughout the Amazon and other tropical regions. Combining high-density, multi-temporal LiDAR data and accurate logging records, we developed an integrated approach to both classify and assess logging impacts in the vertical structure and dynamics of logged forests. We quantified differential recovery periods for understory and overstory vegetation layers. Understory structure resembled its original state within 4-5 years while differences in canopy structure were still detected 8 years following logging. Moreover, we demonstrated an association of post-logging canopy changes with distance from previous understory damage that may be attributed reasonably to post-logging mortality previously identified in field studies. With a simple biomass model based on top-of-canopy height, we estimated explicit biomass decreases due to logging activities and slight biomass gains in areas logged prior to LiDAR acquisitions. We demonstrated that simple metrics derived from the skewness and kurtosis of the LiDAR point clouds can be used to accurately classify logged versus unlogged forests.

Evaluations of compliance to forest management plans can greatly benefit from routine monitoring of logged forests with airborne LiDAR and the impact indicators we developed in this study. Our indicators are objective, quantifiable, and reproducible, and provide objective means for assessing the ecological impacts of selective logging on both carbon stocks and forest structure.

**Author Contributions:** Conceptualization, E.R.P., M.K., and P.D.; Data curation, E.R.P. and M.N.d.-S.; Formal analysis, E.R.P., M.K., P.D., M.L., and D.C.M.; Funding acquisition, M.K.; Investigation, E.R.P., P.D., M.L., M.N.d.-S., and D.C.M.; Methodology, E.R.P., M.K., P.D., and D.C.M.; Project administration, M.K.; Validation, E.R.P., M.K., P.D., and D.C.M.; Visualization, E.R.P. and P.D.; Writing—original draft, E.R.P., M.K., and P.D.; Writing—review & editing, all authors.

**Funding:** Financial support was provided by the Brazilian National Council for Scientific and Technological Development (CNPq, grant 457927/2013-5, E. R. Pinagé) and Science Without Borders program (D. Morton), and the São Paulo State Research Foundation (FAPESP, grant 2015/07227-6, M. Longo). M. Longo was partially supported by an appointment to the NASA Postdoctoral Program at the Jet Propulsion Laboratory, California Institute of Technology, administered by Universities Space Research Association under contract with NASA. M. Keller was supported as part of the Next Generation Ecosystem Experiments-Tropics, funded by the US Department of Energy, Office of Science, Office of Biological and Environmental Research. Financial and administrative support for LiDAR data acquisition and M.N. dos-Santos was provided by the Sustainable Landscapes Brazil project, a collaboration of the Brazilian Agricultural Research Corporation (EMBRAPA), the US Forest Service, USAID, and the US Department of State.

**Acknowledgments:** We acknowledge the support of IFT (Instituto Florestal Tropical) in providing the logging information for Fazenda Cauaxi (shapefiles for logging plots with year of logging, and forest inventory).

**Conflicts of Interest:** The authors declare no conflict of interest.

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
