# Peer review of "Long-Term Impacts of Selective Logging on Amazon Forest Dynamics from Multi-Temporal Airborne LiDAR"

_remotesensing, doi:10.3390/rs11060709_

Round 1
Reviewer 1 Report
The manuscript entitled “Long-term impacts of selective logging on Amazon forest dynamics from multi-temporal airborne lidar” represents interesting original research.
Authors research the potential of ALS (airborne laser scanning, LiDAR) to estimate 3D changes in the forest canopy and understory structure and aboveground biomass following reduced-impact selective logging. The study area is Eastern Brazilian Amazon. The idea of the research is interesting and will attract an audience in the field of airborne LiDAR data acquisition, as well as forest inventory. The manuscript title and abstract are written correct and concise. In the entire manuscript, authors use a standard technical and scientific terminology. After Introduction, the authors explained Methods, as well as achieved results. The scientific experiments and results were conducted according to the scientifically correct approach. The conclusions are logical and based on the results of the research. The manuscript topics fit in Remote sensing aims and scope, especially in LiDAR and laser scanning, as well as Remote sensing applications.
While the topic of the research is new and interesting, the manuscript can be improved. I recommend this manuscript can be accepted after minor revision.
Comments for authors:
1. In the manuscript, authors are inadequately dealing with the problem of the spatial accuracy of the acquired LiDAR data, as well as LiDAR-based CHM and DTM data. Suggest giving as more numbers, data or provide additional accuracy assessment tests. Suggest to research and include in the manuscript literature about the topic mentioned above as: “The Accuracy Assessment of DTM Generated from LIDAR Data for Forest Area – a Case Study for Scots Pine Stands in Poland”, “Accuracy Assessment of Digital Terrain Models of Lowland Pedunculate Oak Forests Derived from Airborne Laser Scanning and Photogrammetry”, “Accuracy of a high-resolution lidar terrain model under a conifer forest canopy ”, “The Evaluation of Photogrammetry-Based DSM from Low- Cost UAV by LiDAR-Based DSM”.
2. Use MDPI standard font (Palatino Linotype) on figures if you can. Please, increase the font size and quality of all figures. Please remove border around figures.
3. About 8% of males have a difficult time distinguishing red from green. Color choice for some figures is not color-blind friendly. Do not use green and red together. So Figures 2, 3, and 8 could be made more accessible. Consult http://colorbrewer2.org or similar websites to discover “colorblind friendly” palettes.
4. Please introduce abbreviations if you want to use it in the further manuscript text or in the abstract (e.g. DTM, DBH, LiDAR, GEDI etc.).
5. In the manuscript text has some typos (e.g. line 31 – 5m -> 5 m; line 50 – “16] [17” -> “16,17]”; line 63 – “30m” -> “30 m”; line 127 – “> 50” -> “>50”; line 152 – “< 42.5” -> ”<42.5”; line 155 – “> 4” -> “>4”; line 183 – “al” -> “al.”; line 178 – “> 10” -> “>10”; line 291 – “0.25ha” -> “0.25 ha”; lines 328-329 – set font to MDPI; lines 370-375 – add space between value and unit (“m”); line 155 – “166m” -> “166 m”, etc.).
6. Please use LiDAR instead of lidar in the entire manuscript.
7. Please expand the Conclusions, summarize main results.
8. Suggest renaming section Methods to Material and Methods.
9. Please, double check all references and reference style.
Author Response
Dear Reviewer,
Thank you for your effort to review our paper. Please find attached a letter with individual responses to each of your comments.
Best regards,
The authors.

Reviewer 2 Report
In this paper, the authors develop a statistical model using airborne Lidar and logging data to to estimate selective logging and to assess related forest degradation and biomass dynamics.
The paper is well written, the methods are solid and correct and the conclusions are supported by the results.
The statistical model is cross validated with independent data and statistical significance is evaluated.
The authors refer to recent relevant research related to the objectives of their study.
În my opinion the whole method is original and relevant for that purpose and may be considered for publication after only minor corrections.
Minor comments:
Line 77 – ‘…Kent, Lindsell, Laurin, Valentini and Coomes [10]’ should be ‘…Ke.t et al. [10]’
The same should be done elsewhere, for example in Line 79 - Kalimantan, Ellis, et al. [35] and line Line 83 - Kalimantan, Melendy, et al. [37]
Author Response
Dear Reviewer,
Thank you for your effort to review our paper, very much appreciated. We carried out a thorough review of references and reference style that included the minor issues you pointed.
Best regards,
The authors.
Reviewer 3 Report
The paper is in my opinion well structured and clear. The aspects treated have been thoroughly investigated.
In the attached text there are some general comments for each chapter and some minor detailed comments.
A small comment could be added in the conclusions regarding the limit of non-ground control of the results obtained. This should also be highlighted with a view to validating the proposed analysis system (it can be done in a subsequent paper maybe in a smaller area).

Author Response

(The authors gave the same response as above.)
